# Electrochemical Corrosion Characterization of Submicron WC-12Co Coatings Produced by CGS and HVAF Compared with Sintered Bulks

Núria Cinca [1,*], Olivier Lavigne [1] , Riberto Nunes Peres [2], Susan Conze [3] , Soeren Hoehn [3], Sergi Dosta [4], Heli Koivuluoto [5] , Chung Kim [6], Fernando Santos da Silva [7], Ville Matikainen [8], Reza Jafari [5] , Elena Tarrés [1] and Assis Vicente Benedetti [2]

1 Hyperion Materials & Technologies, 08107 Barcelona, Spain; olivier.lavigne@hyperionmt.com (O.L.); elena.tarres@hyperionmt.com (E.T.)
2 Instituto de Química, University Estadual Paulista—UNESP, Araraquara 14800-900, SP, Brazil; riberto.nunes@unesp.br (R.N.P.); assis.v.benedetti@unesp.br (A.V.B.)
3 Fraunhofer-Institute for Ceramic Technologies and Systems IKTS, 01277 Dresden, Germany; susan.conze@ikts.fraunhofer.de (S.C.); soeren.hoehn@ikts.fraunhofer.de (S.H.)
4 Departament de Ciència dels Materials i Química Física, Universitat de Barcelona, 08028 Barcelona, Spain; sdosta@ub.edu
5 Materials Science and Environmental Engineering, Faculty of Engineering and Natural Sciences, Tampere University, 33720 Tampere, Finland; heli.koivuluoto@tuni.fi (H.K.); reza.jafari@tuni.fi (R.J.)
6 Hyperion Materials & Technologies, Worthington, OH 43229, USA; chung.kim@hyperionmt.com
7 Instituto Federal de Mato Grosso, Campus Juína, Juína 78320-000, MT, Brazil; silva.fernando@ifmt.edu.br
8 Valmet Technologies Oy, 40700 Jyväskylä, Finland; ville.matikainen@valmet.com
* Correspondence: nuria.cinca@hyperionmt.com

**Abstract:** The electrochemical corrosion performance of WC-12 wt% Co in coating and bulk forms has been evaluated in a 3.56 wt% NaCl solution. The coatings were deposited by means of thermal spray techniques, i.e., cold gas spraying (CGS) and high-velocity air fuel (HVAF) spraying, while bulks with different WC sizes were manufactured by conventional pressing and sintering. Microstructural characterizations and phase composition determinations were carried out using scanning electron microscopy and X-ray diffraction. Differences in WC grain size and morphology, carbide dissolution, and cobalt binder phase transformation are discussed according to the inherent characteristics of each processing method. Together with surface roughness (polished/as-sprayed), these features have been observed to directly affect the electrochemical corrosion performance. Electrochemical measurements (open circuit potential, polarization resistance, electrochemical impedance spectroscopy, and polarization curves) showed that the as-sprayed CGS coating presented an electrochemical behavior similar to those of the bulk materials. This was attributed to the higher metallic character of this coating in comparison to that of the HVAF coating. The polished HVAF coating showed anodic activity lower than those of the bulk samples, most likely due to the presence of cobalt–tungsten carbide phases and eventually the lower amount of Co available for dissolution. Finally, the as-sprayed HVAF coating showed very high resistivity due to the presence of surface oxides generated during the deposition process.

**Keywords:** cemented carbides; thermal spray coatings; corrosion; cold gas spray; high-velocity air fuel

## 1. Introduction

The use of cemented carbides, both as bulks and coatings, is very well extended in industries where high wear resistance is demanded. Their use as coatings, typically by using thermal spray technologies, started in the middle of the last century and created ways to overcome certain geometrical limitations and the high densities of sintered products.

The thermal spray technologies differ in the energy source used to deposit the raw material, i.e., combustion, electric energy, and kinetic energy; however, they all involve melting and/or the ability to plastically deform the feedstock. Therefore, the nominal composition must contain sufficient volumes of metallic binder to allow for the deformation. This is the reason why commercial WC-Co coatings contain an amount of binder phase in the range of 20–30 vol%, corresponding to 12–17 wt% [1].

Although processing bulks and coatings made of cemented carbides have advanced over time, few connections between the two markets can be found. Both markets currently have their well-established niches and exhibit a promising future through the development of compositions and manufacturing parameters [2–4]. Property maps of cemented carbides have been revised to evaluate the balance of hardness, toughness, and wear [5–9], and the property values for thermal spray coatings tend to the lower ranges due to their heterogeneous microstructure [10]. While cemented carbide properties and applications are quite well correlated with their microstructural features, mainly the amount of binder and WC grain size, there is a certain inconsistency in the coating characteristics, given the numerous effects on carbide dissolution.

The powder metallurgical processing of cemented carbide products involves pore elimination first through a shrinkage step in solid-state sintering, and when the binder is molten, further densification occurs. The final microstructure is highly dependent on the carbon balance within the composition and can be predicted from phase diagrams. By contrast, conventional thermal spray technologies consist of fully or partially melting of particles inside or outside a spray gun and a rapid cooling afterward upon impingement onto the substrate, thus leading to microstructural phases that cannot be directly predicted under equilibrium conditions. Conventional thermal spray microstructures may also contain some level of porosity and oxidation, depending on particle velocity and temperature conditions.

The most widely applied spraying technology in the market for cemented carbides is high-velocity oxy fuel (HVOF), which became the industry standard technology for cemented carbide coatings, typically in the thickness range of 100 to 500 μm [11]. In order to reduce decarburization, porosity, and oxidation, cold gas spraying (CGS) attempts have been performed during the last years [12–15]. CGS would then provide a composition closer to that of a bulk carbide obtained by pressing and sintering. However, this technique can still create low deposition efficiencies for cemented carbides. In between HVOF and CGS, high-velocity air fuel (HVAF), which uses air instead of oxygen for combustion, is being considered to overcome the shortcomings of the other two coating methods. HVAF spraying is a 'warm spray' process that is cooler than that of HVOF but hotter than cold spraying, reaching spray rates even higher than that of HVOF and with improved hardness properties.

Many times, to evaluate the heterogeneity of coating microstructures, electrochemical corrosion testing is used. According to the level of electrolyte penetration, one can determine the presence of internal defects. The corrosion resistance is higher the lower the cobalt binder content [10] and the higher the coating compaction process. Studies comparing different types of WC-Co coatings deposited with high-velocity oxygen fuel (HVOF), high-velocity air fuel (HVAF), and cold gas spray (CGS) indicated that the corrosion results are highly influenced by the spraying technique [16–19]. In general, the corrosion resistance followed the order: CGS > HVAF > HVOF.

The comparison of bulks with HVAF and CGS coatings appears to be quite appealing. Some authors have attempted to use thermal spraying in applications where bulk cemented carbides are typically used [19], but a direct comparison of properties of bulk cemented carbides and coatings is found very rarely in the literature [12,20–22]. Therefore, the present work is first aimed at comparing the primary microstructural features of WC-12Co bulks and coatings. Figure 1 shows a fishbone of the corresponding two coating processes and the conventional pressing and sintering manufacturing, illustrating the large number of variables that influence the final microstructures. The second goal of this work is to

compare the electrochemical corrosion performance of WC-12Co coatings obtained by different techniques; while the tribology performance has been previously compared and reviewed [10], no such studies exist on the corrosion performance.

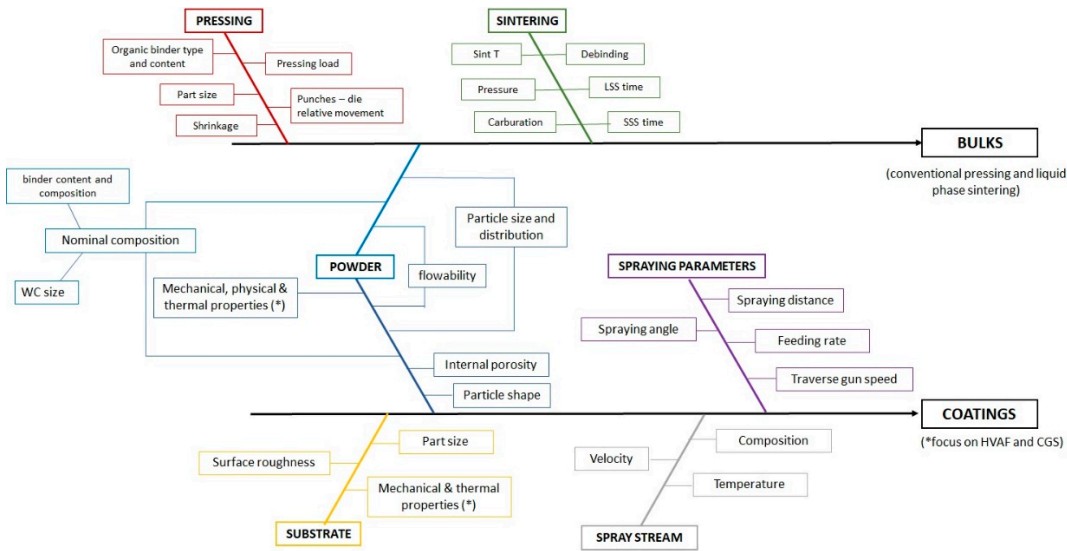

**Figure 1.** Fishbone analysis of variables affecting production of cemented carbide bulk and coatings. The factors with an asterisc (*) may be critical for HVAF and CGS spraying but may not be that much for other thermal spraying techniques.

## 2. Materials and Methods

### 2.1. Feedstock and Sample Preparation

The powders used for CGS and HVAF deposition were WC-12 wt%Co cermets (WC-12Co) obtained by agglomeration (via spray drying) and sintering (supplied by Fujimi Inc., Kiyosu, Japan, and Durum Verschleißschutz GmbH, Willich, Germany, respectively). Both were characterized as containing submicronic WC particles. Figure 2 shows the microstructure of the two powders. Both exhibited a spherical morphology and porous microstructure. The average level of porosity in all particles appears to be more uniform in the case of the HVAF powders, while the porosity in CGS powders differs more from one particle to another. They were deposited onto AA 7075-T6 alloy (0.18 wt% Cr, 1.2 wt% Cu, 2.1 wt% Mg, 5.1 wt% Zn, and balance Al) substrates in order to have a better deposition efficiency of the first layer of particles by CGS spraying through particle embedment within this softer substrate. For CGS spraying, the substrates were degreased with acetone, and the surfaces were abraded with P240 SiC paper, resulting in a surface roughness (Ra) of ~1 μm. For HVAF spraying, the substrates were only cleaned with acetone prior to spraying.

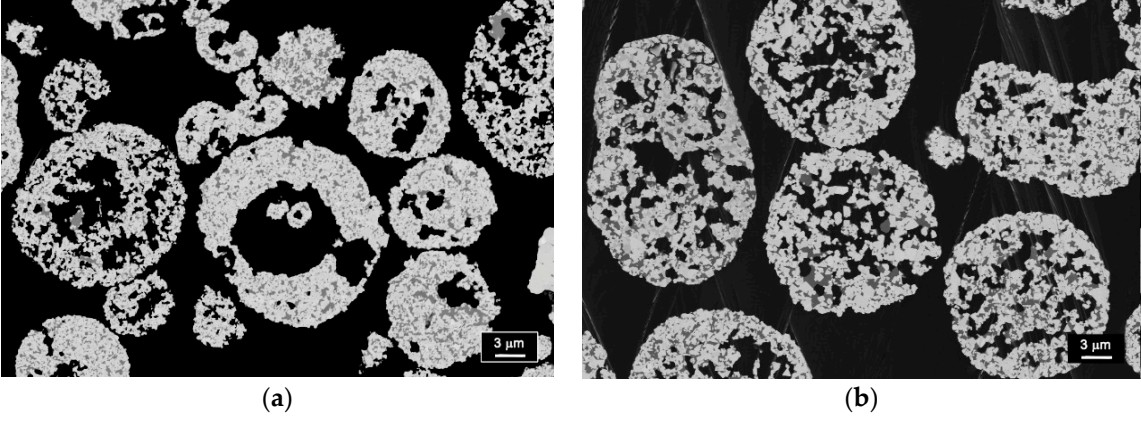

(**a**)        (**b**)

**Figure 2.** Cross section microstructures of WC-12Co powders for spraying (**a**) CGS and (**b**) HVAF coatings.

The spraying of the WC-12Co CGS coatings was performed by using a Kinetics 4000/17 kW system (Cold Gas Technology, Ampfing, Germany) operating at a maximum operating gas pressure of 40 bar and gas temperature of 800 °C, and with nitrogen as the carrier gas. A WC-based nozzle (D24) was used to deposit the coatings. The parameters were optimized to improve the deposition efficiency and adhesion [19].

Thick (~250 μm) and thin (~50 μm) HVAF coatings were produced by using an M3 HVAF spray torch (Uniquecoat Technologies, Oilville, VA, USA) with propane and compressed air as the process gases. A long combustion chamber and a convergent–divergent 4L2 nozzle were used to achieve efficient melting and high particle velocities [23]. The purpose of having two different coating thicknesses was to provide a corrosion comparison of the thin coating with the CGS coating, which is also a thin coating due to its lower deposition efficiency, and the thick coating was intended to be polished and directly compared with bulks having the same surface preparation. Coating porosity was evaluated by means of image analysis with the ImageJ software.

The powders used to produce bulk parts were also WC-12 wt%Co, prepared by agglomeration. Three different powders were used to obtain ultrafine (12CoUF), fine (12CoF), and medium (12CoM) WC grain size particles through pressing and liquid phase sintering (sinter-HIP process). After sintering, materials did not show any undesirable phases (graphite or eta phase).

## 2.2. Structural, Morphological, and Chemical Characterization

Powders, coatings, and bulks were characterized by scanning electron microscopy (SEM) using a Carl Zeiss Crossbeam 550 microscope coupled to an X-ray microanalysis (EDS) system. Cross-sections were prepared by grinding and polishing in steps using diamond suspensions, ultimately down to the final polish with a 1 μm diamond suspension.

The phase composition was analyzed by X-ray diffraction (XRD) using a Bruker (Billerica, MA, USA) D8 FOCUS diffractometer with an LYNXEYE detector. High-resolution scans with step sizes of 0.01° were carried out at the region of 35–50° for the better identification of cobalt peaks and other secondary phases. The porosities of the coatings were estimated by analyzing the images using Image J software (version 1.41o), according to the ASTM E2109-01 protocol.

Additionally, several microstructural parameters were characterized, i.e., WC average grain size ($d_{carbide}$, in μm), binder volume fraction ($V_{binder}$), carbide contiguity ($C_{carbide}$), and binder mean free path ($\lambda$). The $d_{carbide}$ was measured by the linear intercept technique following the ISO 4499-2 standard.

The $C_{carbide}$, which corresponds to the surface shared by a carbide with another carbide, was determined according to the empirical relation shown in Equation (1), which considers the simultaneous effect of binder content and carbide mean grain size based on an extensive data collection from open literature [24].

$$C_{carbide} = 0.036 + 0.973 \times exp\left(\frac{-d_{carbide}}{3.901}\right) \times exp\left(\frac{-V_{binder}}{0.249}\right) \tag{1}$$

The binder mean free path is calculated using the true mean free patch definition [25]:

$$\lambda_{binder} = \frac{1}{1 - C_{carbide}} \times \frac{V_{binder}}{V_{carbide}} d_{carbide} \tag{2}$$

where $V_{carbide}$ is the carbide volume content.

The use of these formulas to obtain the values for the as-sprayed coatings is an approximation since, although the powders are in sintered form, the afterward spraying involves further rearrangement of WC particles.

### 2.3. Electrochemical Corrosion

For the sake of comparison on the surface state, the polished (down to 1 μm diamond suspension) bulk samples were also compared with the polished HVAF coating, while the as-sprayed HVAF and as-sprayed CGS coatings were compared as well. Any likely influence on the surface nature and morphology could therefore be detected. This procedure was followed because of the more limited deposition efficiency of the CGS coating. Table 1 presents the details of the studied samples.

**Table 1.** Surface conditions of the bulks and coatings' thicknesses used in the electrochemical studies.

| Title 1 | Surface State | Coating Thickness (μm) | Coating Roughness (Ra, μm) |
|---|---|---|---|
| CGS | as-sprayed | ~50 | 4.7 ± 0.3 |
| HVAF | as-sprayed | ~50 | 2.2 ± 0.1 |
| HVAF | polished | ~250 | N/A |
| 12CoM | polished | N/A | N/A |
| 12CoF | polished | N/A | N/A |
| 12CoUF | polished | N/A | N/A |

Corrosion measurements were conducted with a Gamry 1010E interface (Gamry Instruments, Warminster, PA, USA) at $25 \pm 1$ °C in an aerated solution containing 3.56 wt% of NaCl. A conventional three-electrode setup was used, with a graphite counter electrode and a saturated calomel electrode (SCE) as reference. The working electrode (samples with a polished surface down to 1 μm diamond suspension) was placed in a Teflon holder, leaving an exposed surface area to the electrolyte of 0.785 cm$^2$. The open-circuit potential (OCP) was recorded for 24 h and 3 h for coated and bulk samples, respectively. These durations ensured OCP steady state before polarization resistance (Rp), electrochemical impedance (EIS), and potentiodynamic measurements. Rp measurements were conducted at $+/-10$ mV around OCP at a scanning rate of 0.167 mV/s. The EIS measurements were performed after 24 h of immersion in the electrolyte for the coatings and 3 h for the bulk materials from 100 kHz to 10 mHz by applying a sinusoidal potential perturbation of 10 mV rms on the open circuit potential and collecting 10 points/frequency decade. For all samples analyzed by equivalent electric circuit (EEC) theory, the data were validated using the Kramers–Kronig transform (KKT) software available in the Gamry system software. The EEC models obtained using Z-view software were used for quantitative analysis of the EIS responses. After EIS measurements, the potentiodynamic curves were obtained from $-0.150$ V$_{OCP}$ to 0.8 V$_{SCE}$ at 0.167 mV/s, but only represented up to 0.0 V$_{SCE}$.

For each set of electrochemical experiments, at least two specimens of each material, prepared in the same batch, were used. For each specimen, the experiments were performed in the same area in the following sequence: open circuit potential (OCP), three polarization resistance (Rp), followed by two consecutively electrochemical impedance measurements, and then one potentiodynamic polarization curve.

### 3. Results

#### 3.1. Morphology and Structural Composition of CGS Coatings, HVAF Coatings, and Bulks

Figure 3 shows the cross-sections of CGS and HVAF coatings onto the Al alloy. With both techniques, which produce high particle velocities, the large substrate deformation upon particle impingement is easily visible, with the round morphology of the particles at the coating–substrate interface. The much softer Al substrate leads, in both cases, to particle penetration resulting in a uniform layer produced by the first incoming particles.

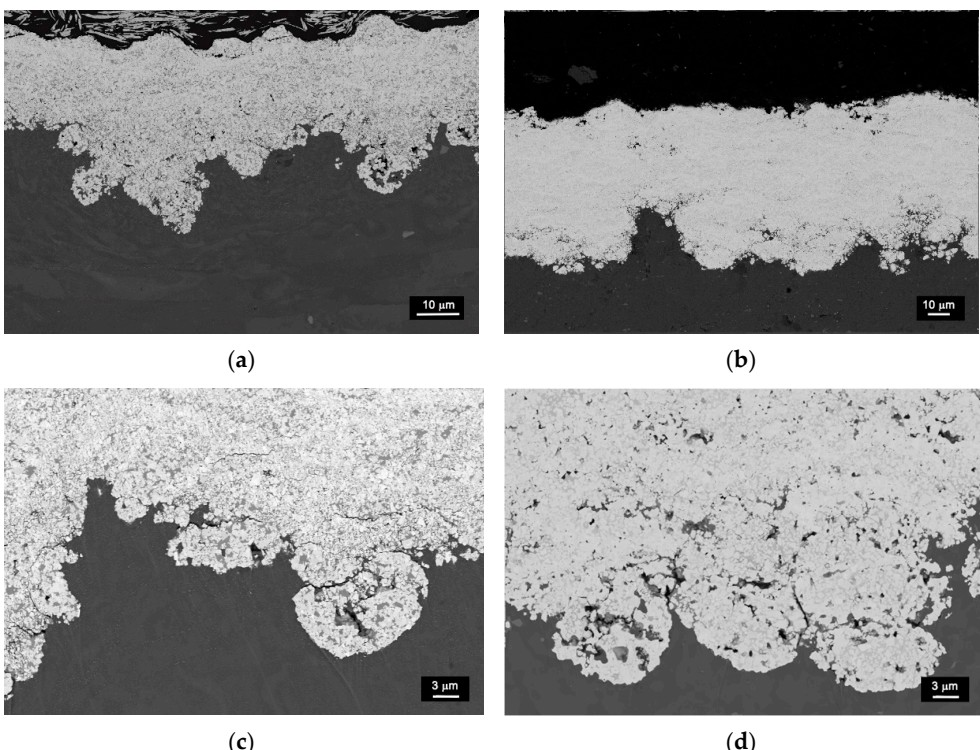

**Figure 3.** Cross section of (**a**) CGS and (**b**) HVAF WC-12Co coatings, and (**c**,**d**) respective coating–substrate interfaces.

Figure 4 shows details of the microstructure of the two WC-12Co coatings. The CGS coating shows nano-sized and some submicronic WC particles that were already observed in the feedstock, whereas the HVAF coating consists of more micron and submicron WC particles. The high impact of the CGS powder particles in a solid state is observed to lead in part to the cracking of the WC grains, see Figure 4c, and to the displacement of the WC and Co network. Such cracking is not observed in the HVAF coating, which primarily exhibits porosity and carbide dissolution, resulting in a rounder WC morphology (Figure 4d). By contrast, Figure 5 shows the uniform microstructure of the bulk pressed and sintered samples with the typical prismatic shape of WC grains.

Table 2 presents the WC size measurements together with contiguity and binder mean free path values. The smallest WC grain size corresponds to the CGS sample, followed by the 12CoUF bulk and then the HVAF coating. Furthermore, the values of the coating porosity are given, with the value for HVAF coatings being larger than that for CGS coatings. This is not surprising given the easier gas entrapment in those processes involving some particle melting.

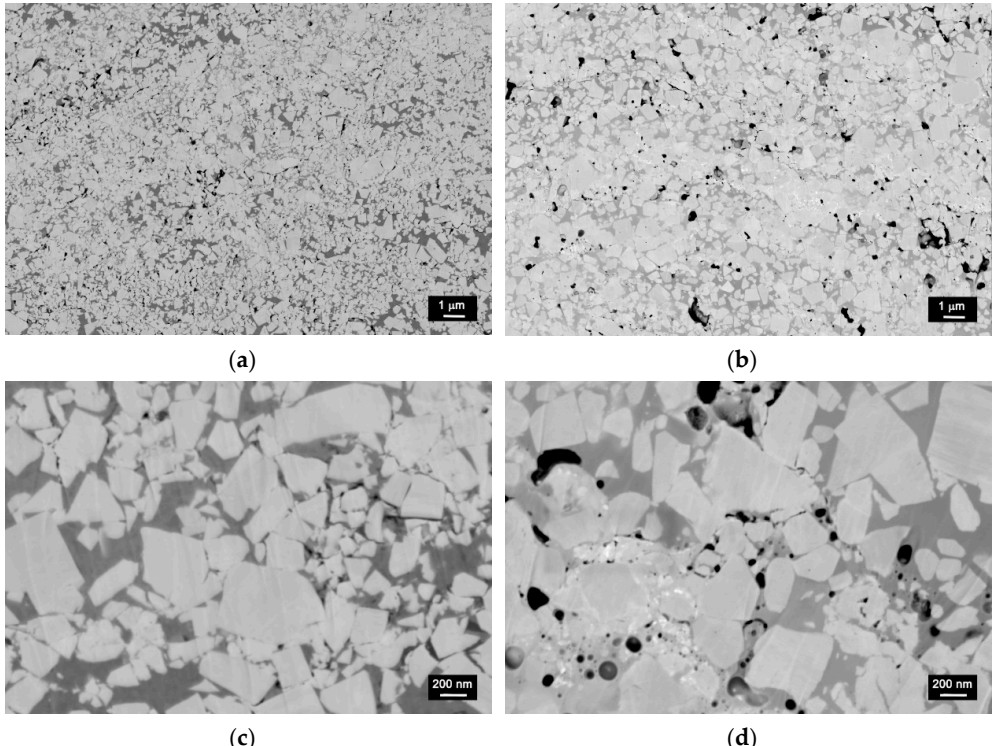

**Figure 4.** Microstructure of (**a**,**c**) CGS and (**b**,**d**) HVAF WC-12Co coatings.

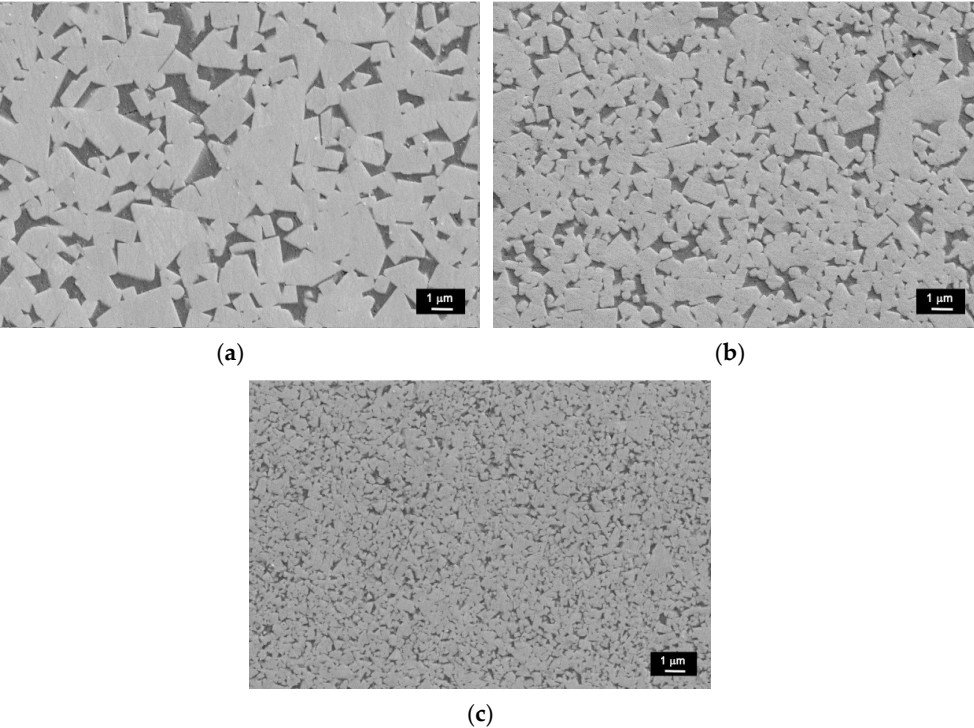

**Figure 5.** Microstructure of the bulk (**a**) 12CoM, (**b**) 12CoF and (**c**) 12CoUF samples.

**Table 2.** Microstructural parameters of the examined samples.

| Sample | Average WC Size (µm) | Contiguity | Mean Free Path (µm) | Coating Porosity (%) |
| --- | --- | --- | --- | --- |
| CGS | 0.14 | 0.46 | 0.06 | 0.7 ± 0.1 |
| HVAF | 0.34 | 0.44 | 0.15 | 4.2 ± 0.3 |
| 12CoM | 0.98 | 0.38 | 0.38 | N/A |
| 12CoF | 0.60 | 0.41 | 0.25 | N/A |
| 12CoUF | 0.27 | 0.45 | 0.12 | N/A |

The XRD spectra presented in Figure 6 identify the phase composition of several samples. Peak broadening and carbide dissolution are observed in the HVAF coating, while in the CGS coating, no carbon loss has taken place. The present identification of the HVAF coating cannot exclude the presence of any other phase since peaks tend to be broadened, convoluted, or shifted into position [21]. When sprayed by CGS, no secondary phases can be primarily observed, but the broadening of peaks can be indicative of the accumulated strain upon particle impingement. This can be different from the HVAF coating, where broadening might be rather attributed to nanocrystalline phases and some amorphization as a result of non-equilibrium phenomena. However, the latter could not be confirmed.

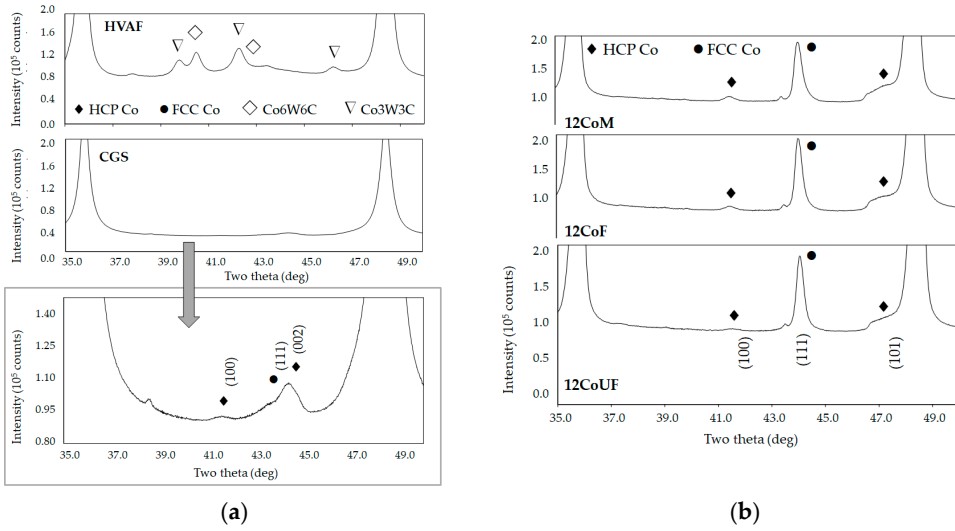

(a)  (b)

**Figure 6.** X-ray diffraction spectra (**a**) coatings and (**b**) bulk materials. Indexed patterns: PDF-ICDD-05-0727 for Co-HCP and 015-0806 for Co-FCC.

Interestingly, the CGS coating contains cobalt in the HCP phase, whereas the powder contains cobalt in the FCC phase coming from the sintered powders. Thus, due to the impact, the Co FCC to HCP transformation occurred. The three bulks primarily exhibit the FCC phase cobalt. The small peak at 43.5° could not be properly identified according to the X-ray database. This cannot exclude the occurrence of some $WC_{1-x}$ phases.

## 3.2. Electrochemical Corrosion

Figure 7 shows the potentiodynamic polarization curves obtained after stabilizing the open circuit potential in 3.56 wt% NaCl solution at 25 °C:

- 24 h for as-sprayed coatings and polished HVAF;
- 3 h for polished 12CoUF, 12CoF, and 12CoM bulk materials (see Table 1).

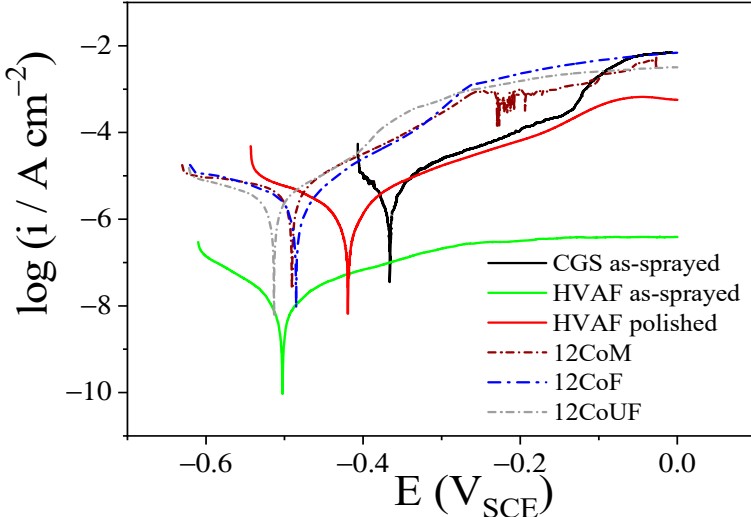

**Figure 7.** Potentiodynamic polarization curves for as-sprayed coatings and polished samples obtained in aerated and unstirred 3.56 wt% NaCl solution at 0.167 mV/s and 25 °C.

The curves for the coatings are in the positive potential direction, while those for the bulk materials are more towards the negative one. Between the two coatings, the CGS coating sample exhibits less negative corrosion potential, which suggests a lesser tendency to dissolve in this medium (see also values in Table 3). For the as-sprayed HVAF coating sample, the corrosion potential is ~−0.50 $V_{SCE}$, which is around ~70 mV more negative than for the polished HVAF. This suggests that the different microstructures, surface compositions, and roughnesses can be responsible for this observed potential difference. The as-sprayed CGS coating sample presents the highest corrosion potential with respect to the other samples. CGS coatings are typically almost oxide-free [15], but due to the porosity of the top layer that is more active, some oxide can be formed in contact with the electrolyte. Table 3 also shows that the 12CoUF sample, which has the smallest WC grain size among the bulk materials, presents the lowest corrosion potential among the polished samples.

**Table 3.** Corrosion potential ($E_{corr}$) and polarization resistance ($R_p$) for the different samples in 3.56 wt% NaCl solution at 25 °C.

| Sample | $E_{corr}$ (V vs. SCE) | $R_p$ (kΩ cm²) |
|---|---|---|
| CGS as-sprayed | −0.367 | 2.8 ± 0.3 |
| HVAF as-sprayed | −0.504 | 889 ± 50 |
| HVAF polished | −0.420 | 7.2 ± 0.4 |
| 12CoM | −0.489 | 3.1 ± 0.3 |
| 12CoF | −0.468 | 3.4 ± 0.3 |
| 12CoUF | −0.510 | 2.7 ± 0.3 |

Observing the cathodic branch, one can see that almost all curves present a tendency to achieve a limiting current at more negative potentials, whereas the anodic branch shows the current density increasing continuously. No surface passivation is observed; on the contrary, a limiting current density, $i > 5$ mA cm$^{-2}$, was measured (not shown) for all samples, except for the HVAF sample, at potentials higher than 0.10 $V_{SCE}$. The formation of a passive layer on cobalt in a chloride-neutral medium is impeded by the aggressiveness of chloride.

The highly complex nature of coatings due to their heterogeneous microstructure makes the quantitative analysis of the potentiodynamic polarization curves difficult. There-

fore, the polarization resistance obtained from the linear polarization curves near the zero current could help the comparison among the samples. The highest polarization resistance (889 k$\Omega$ cm$^2$) was measured for the as-sprayed HVAF coating and, consequently, the lowest corrosion current density among the studied samples (Table 3). This extremely high resistance suggests that the cobalt surface passivated with the formation of a thick oxide layer during the sample preparation, and the 24 h of immersion in chloride solution are not enough to break down the passive surface. For the polished HVAF sample, the polarization resistance decreases to 7 k$\Omega$ cm$^2$, which is still higher than the resistance for all the others. The other samples show almost the same R$_p$ values within the experimental deviation, with CGS and 12CoUF samples indicating a tendency to be more active. The 12CoF and 12CoM samples show almost the same polarization resistance values.

Figures 8 and 9 show electrochemical impedance diagrams (the complex plane) and impedance moduli, and phase angle Bode plots for as-sprayed CGS and HVAF coatings and polished HVAF, 12CoM, 12CoF, and 12CoUF samples obtained in aerated and unstirred 3.56 wt% NaCl solution after 24 h immersion for the coatings (HVAF and CGS) and 3 h for bulk materials. The insert in the Nyquist graphic of Figure 8a is a zoom of the high-frequency region to put in evidence the high impedance of the polished HVAF sample. As can be seen in Figure 8a, the complex plane plots are characterized by an incomplete semicircle from high to low frequencies with some dispersion of the data at very low frequency, encompassing less than one frequency decade for the bulk materials. This dispersion may result from an instability of the system due to the adsorption/desorption of corrosion products. For the bulk materials, the semicircle increases as 12CoUF < 12CoF < 12CoM; however, the 12CoM and 12CoF samples exhibit almost identical impedance. For the polished HVAF sample, the complex plane plot at low frequency is very different from those of the other samples, indicating a continuous increase of the impedance at the low-frequency region. The electrochemical impedance diagrams for the as-sprayed HVAF sample (Figure 9) are much more complex than those for the other samples, presenting a real high impedance at high frequency, which reflects a low conductivity of the surface. In the Nyquist plot of this figure, more than one incomplete semicircle can be seen. One may result from a low conductivity layer, while another at the middle-low frequency range probably is associated with charge transfer resistance and a continuous increase in real impedance at lower frequencies suggesting a high resistance feature. In order to understand the results of this sample, more details should be investigated in the future. Even considering that the experimental data were validated, which means that this is not due to an experimental artifact, and despite the several tested EECs, it was not possible to find an EEC that fitted correctly to the experimental data as for the other samples.

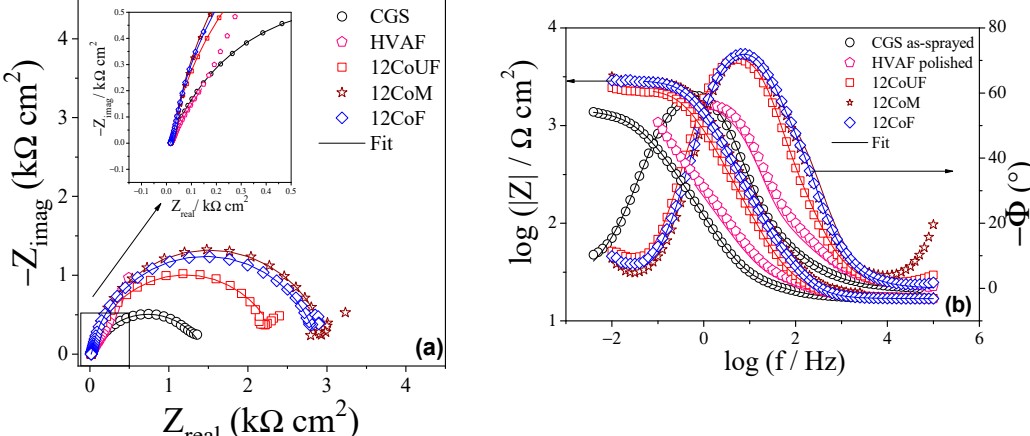

**Figure 8.** (**a**) Complex plane plots and (**b**) impedance moduli and phase angle Bode plots for as-sprayed CGS and polished samples obtained in aerated and unstirred 3.56 wt% NaCl aqueous solution at 25 °C.

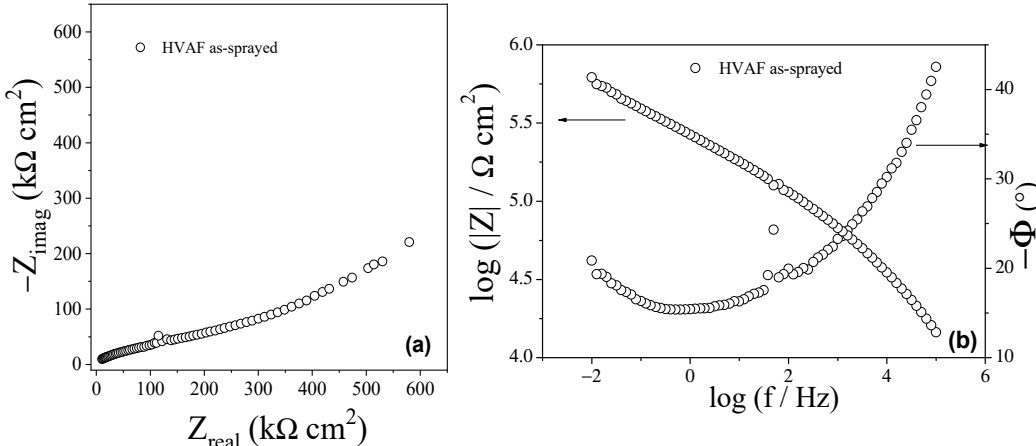

**Figure 9.** (**a**) Complex plane plot and (**b**) impedance moduli and phase angle Bode plot for the as-sprayed HVAF sample obtained in aerated and unstirred 3.56 wt% NaCl aqueous solution at 25 °C.

When observing the phase angle vs. log frequency plots in Figure 8b, two groups of samples can be considered: the as-sprayed CGS and polished HVAF coatings with a maximum at phase angles around −60° (~0.6 Hz) and ~−55° (2 Hz), respectively, and the bulk materials with a maximum at phase angle around −70° (between 5 and 8 Hz), with the 12CoF and 12CoM samples producing practically the same values. These values could indicate that bulk materials have higher capacitive behavior and a lower charge transfer resistance. A closer observation of these curves suggests more than a one-time constant due to the non-symmetry of the curves, which was confirmed by the fitting of the experimental data. The impedance moduli vs. frequency plots in Figure 8b show a tendency to achieve a $|Z|$ value independent of the frequency at low frequencies, with the value of $|Z|$ for as-sprayed CGS coating (~1.4 kΩ cm$^2$), 12CoUF (~2.5 kΩ cm$^2$), 12CoF (~2.9 kΩ cm$^2$), and 12CoM (~3.2 kΩ cm$^2$). The $|Z|$ assumes practically the same values for bulk materials, a little higher when compared to as-sprayed CGS coating. For polished HVAF samples, the $|Z|$ value seems to increase as the frequency decreases, and no tendency to achieve a frequency-independent value is observed, suggesting that another process occurs, for instance, a diffusion inside the coating, requiring more studies to elucidate the process. The graphics in Figure 9b show the impedance modulus $|Z|$ and the phase angle Bode plot for the as-sprayed HVAF coating obtained at the same conditions as for the polished HVAF sample. As observed for the Nyquist plot, the Bode plots are completely different from those obtained for the other samples, being the phase angle decreasing from ~−40° at 100 kHz to around −15° at the middle frequency range, oscillating around this value until 0.1 Hz, clearly indicating that the system is dominated by a highly resistive behavior. This resistive behavior is confirmed by the $|Z|$ vs. frequency in the same figure as $|Z|$ increases continuously from around 15 kΩ cm$^2$ at 100 kHz to 585 kΩ cm$^2$ at 0.1 Hz for the as-sprayed HVAF sample.

Figure 10 shows the SEM micrographs of the polished HVAF and 12CoUF surfaces obtained after the electrochemical experiments. These images demonstrate that the electrochemical experiments affected the HVAF sample surface more than it did the 12CoUF sample surface. This is a result of the dissolution of the cobalt and amorphous phases, which generate holes and pores. This effect is also observed in the bulk materials, resulting in binder leaching around the WC particles due to the preferential dissolution of cobalt.

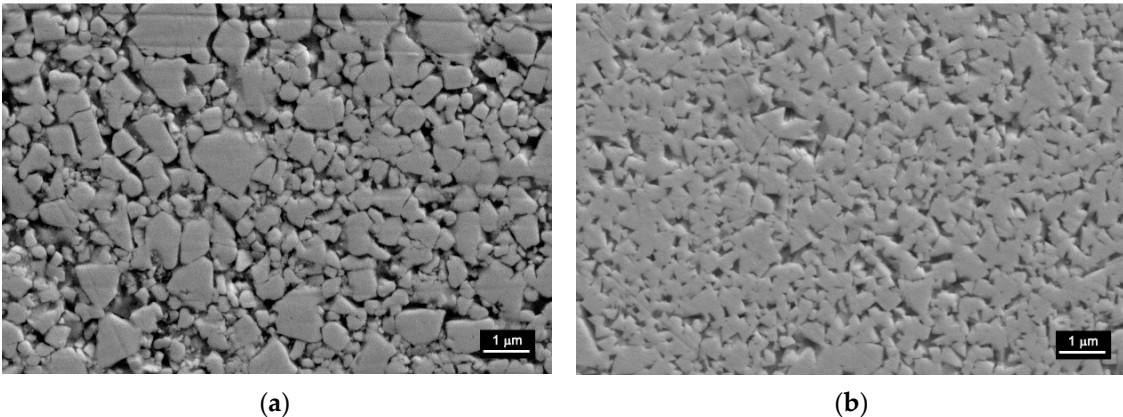

**Figure 10.** SEM images of (**a**) polished HVAF and (**b**) 12CoUF samples after electrochemical tests.

## 4. Discussion

The CGS and HVAF processes used here for the spraying of WC-12Co powders create higher particle velocities and lower spray stream temperatures than other thermal spraying techniques. The high kinetic contribution in both processes implies that the mechanical properties of powder and substrate are key influences in the formation of the first layer and coating buildup. The different combinations of hard–hard, soft–soft, hard–soft, and soft–hard particle–substrate interactions determine the coating adhesion, especially for the solid-state nature of the CGS process [26]. As observed, the much softer Al alloy substrate leads to particle penetration of the hard WC-12Co particles resulting in a uniform layer produced by the first incoming particles. The ability for coating buildup afterward relies more on the plastic deformation of the particles [27]. Successful attempts with 25 wt% and 17 wt%Co have been reached [14,18,28], while 12 wt%Co is more challenging with a higher critical effect of internal porosity [12,29]. The higher the particle porosity, the more deformation the particle accepts on impact resulting in higher compaction and better final deposition efficiencies. Previous authors have reported that there exists a critical WC size so that a good coating density is achieved, therefore, also playing a key role together with internal porosity [13,30].

The characteristics of the CGS process avoid decarburization, while some carbide dissolution is observed in the HVAF coating with the presence of $Co_xW_xC$ (with $x = 3$ and 6). Additional features observed are the WC grain sizes and shapes, which significantly differ in the two processes. The WC grain size reduction in CGS coating is related to carbide fracture due to the high impact of particles, while the one in the HVAF coating depends on the thermal history of the particles within the flame. All this results in a low uniformity of the coatings compared to the bulk samples, where the Ostwald Ripening process drives the coarsening of WC grains during the liquid phase sintering [31].

The above-mentioned features directly affect the corrosion performance of the samples. The potential corrosion values of the coatings are likely attributed to the presence of a cobalt oxide layer formed on the surface by interaction with the electrolyte, a complex microstructure, and roughness, while the lowest value for the 12CoUF bulk sample suggests a higher activity of the Co metallic phase. The electrochemical activity of metals under stress or strain is known to be generally promoted [32]. As the 12CoUF sample had the smallest WC particle size among bulk samples, the tensile residual stresses in the Co phase are expected to be the highest [33,34]. This could promote the dissolution of the Co metallic phase. Moreover, dissolution starts from the center of the metallic islands [35]; because the binder mean free path is the smallest for this sample, the rates of Co dissolution might also be enhanced by the WC cathodic areas surrounding thin Co anodic areas.

Similarities that can be found in the potentiodynamic polarization curves are as follows: (i) the cathodic branches of almost all curves present a tendency to achieve a limiting current at more negative potentials, which agrees with the fact that in this medium

(pH = 6.5) the cathodic reaction is the oxygen reduction occurring on the WC phase [36,37], while (ii) in the anodic branches, the current increases continuously with the potential due to the cobalt dissolution reaction [35,38]. No surface passivation is observed; the formation of a passive layer on cobalt in a chloride neutral medium is impeded by the aggressiveness of chloride. The electrolyte pH is of crucial importance for the WC-Co system as the cobalt matrix is very sensitive to dissolution in acidic media, while WC tends to be dissolved in alkaline media [36]. Then, a high cathodic overpotential can move the pH on the cathodic sites (WC phase) to an alkaline region facilitating the WC dissolution. If a high cathodic potential may lead to a change in the pH at the WC surface, causing its dissolution, a high anodic potential may lead to the formation of holes or pores allowing electrolyte penetration inside the coating laterally and/or deeply, which could limit the mass transport of cobalt cations to the solution, facilitating its hydrolysis and causing local pH decrease [37,39]. The decrease in pH accelerates the cobalt dissolution. For the coatings, further complexity is imposed as a result of the porosity occurrence at inter-splat boundaries, which, together with likely undesired phases, introduce further paths for electrolyte penetration. The WC grain size, binder content, and binder type have been found to have a large influence on the electrochemical behavior of the bulk materials [34,40].

The as-sprayed HVAF coating presents an extremely high polarization resistance value compared to the as-sprayed CGS coating. This suggests that the cobalt surface was passivated with the formation of a thick oxide layer during the sample preparation and that the 24 h immersion in the chloride solution was not enough to break down the passive surface. The lower polarization resistance value of the polished HVAF coating is likely attributed to the presence of remaining oxides, $Co_xW_xC$ phases, and, probably, some level of amorphization of the microstructure [37]. Some EDS analyses (not shown) were performed on the as-sprayed HVAF coating surface just after coating preparation, and some oxygen was detected, which confirms the previous hypothesis. The CGS coating, together with the 12CoUF sample, appeared to be more active; the residual stresses inherent to the CGS technique and the small WC grain size of 12CoUF among the bulk material could be related to the increase in the activity of these samples.

EIS diagrams may reflect the features related to the galvanic coupling between the two phases of the studied samples. At open circuit potential, the exposed cobalt oxidizes with the reduction of dissolved oxygen at the WC sites. As mentioned above, the galvanic coupling with the WC phase may increase the dissolution rate of cobalt present in the samples, and the hydrolysis of the cobalt cations may lead to acidification of the solution, mainly in regions where the access to oxygen, which can cause the corrosion of cobalt to be accelerated, is limited [35,36,41]. This condition can be present in the CGS and polished HVAF coatings for immersion times around or higher than 24 h in chloride-neutral solution. Similar behavior can be expected for the bulk materials when cobalt is dissolved. The formation of pores and/or holes (due to binder leaching) that is accelerated by applying an anodic potential is confirmed in Figure 10. These defects lead to an additional resistance apart from the charge transfer resistance associated with the corrosion process (anodic and cathodic reactions), which accounts for the solution resistance inside the pores.

The EIS data were quantitatively analyzed for all samples, except for the as-sprayed HVAF sample, using the EEC theory. The equivalent electric circuits were chosen, taking into account the structures of the surface samples, the best fitting of the experimental data indicated by a low value of $\chi^2$, and the lowest residual error of each parameter of the equivalent circuit. In the case of the as-sprayed HVAF sample (EIS diagram of Figure 9), the EEC suggested for the other samples cannot represent a better model for this one, and more studies are necessary to interpret its EIS response. The equivalent circuit used here is based on the following:

- For all analyzed samples using EEC, the amount of oxides present on the surface is low, there is no passivation of cobalt, and the global process is dominated by the cobalt dissolution and oxygen reduction;

- The metallic regions are more active and can be dissolved faster, generating holes, pits, and pores;
- Oxygen is primarily reduced at the surface of the samples, where its access is easier;
- Due to the relatively short immersion time, the electrolyte did not reach the substrate of the coatings as suggested by the potential corrosion values (Table 3);
- For polished HVAF samples, some remaining oxides would change the impedance response at frequencies below 0.1 Hz, but the same EEC (Figure 11) to as-sprayed CGS coating and bulk materials may explain the impedance results of polished HVAF samples at high- and middle-frequency ranges.

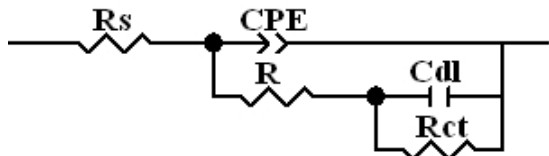

**Figure 11.** Equivalent electric circuit used to adjust to the experimental data of EIS.

Figure 11 depicts the EEC used that adjusts to the EIS experimental data for as-sprayed CGS and polished HVAF coatings, as well as bulk materials. The values of each element of the equivalent circuit with the respective error are presented in Table 4.

**Table 4.** Parameters of the EEC used to fit to the EIS experimental of different samples. Rs was normalized to the lower value measured for the samples (Rs = 16 $\Omega$ cm$^2$) to better represent all curves at the same graphic, and therefore, the Rs value was not added to the table.

| Conditions \ Element of EEC | As-Sprayed CGS | Polished HVAF | Polished 12CoUF | Polished 12CoF | Polished 12CoM |
|---|---|---|---|---|---|
| CPE-T (mF cm$^{-2}$ s$^{(n-1)}$) | 1.42 (2.3) * | 1.48 (0.6) | 0.166 (1.0) | 0.104 (0.75) | 0.103 (0.6) |
| n | 0.74 (0.3) | 0.66 (0.2) | 0.87 (0.2) | 0.89 (0.1) | 0.87 (0.1) |
| R ($\Omega$ cm$^2$) | 18 (5.2) | 47 (3.0) | 74 (8.9) | 163 (4.3) | 574 (4.3) |
| CPE$_{dl}$-T ($\mu$F cm$^{-2}$ s$^{(n-1)}$) | 650 (4.8) | - | - | - | - |
| n$_{dl}$ | 0.88 (0.9) | - | - | - | - |
| C$_{dl}$ ($\mu$F cm$^{-2}$) | - | 101 (2.1) | 27.4 (4.3) | 16.6 (2.5) | 11.6 (3.2) |
| R$_{ct}$ (k$\Omega$ cm$^2$) | 1.4 (0.3) | 4.3 (9.2) | 2.3 (0.6) | 2.7 (0.4) | 3.4 (0.7) |
| $\chi^2$ ($10^{-4}$) | 2 | 2 | 8 | 3 | 3 |

\* The values in parenthesis represent the error of each element of the equivalent circuit. The s is the complex frequency and in the frequency domain s $\equiv$ jw.

The circuit (Figure 11) is composed of the solution resistance (Rs) in series with two cascade subcircuits: R//CPE and R$_{ct}$//C$_{dl}$. The CPE//R subcircuit encompasses all contributions arising from the global surface: resistances and capacitances associated with the charge transfer process at the surface, contributions from an existing incipient film, or a film related to the accumulation of corrosion products on the surface. R$_{ct}$/C$_{dl}$ is related to the charge transfer resistance (R$_{ct}$) of the electrochemical process inside the pores, holes, or pits, and C$_{dl}$ is the capacitance of the electrical double layer. The CPE is the constant phase element and substitutes the capacitance of a non-ideal system and is composed of the admittance CPE-T and n, the exponent, both independent of the frequency. The n assumes values equal 1 for a pure or ideal capacitor, zero for a pure resistance, 0.5 for a diffusion process, $-1$ for an ideal inductor, and 0.5 < n < 1 for heterogeneous current distribution on the sample surface due to the presence of different phases, microstructures, roughness, etc. [42].

The admittance values (CPE-T) are around one order of magnitude higher for the coatings than for those of the bulk materials, suggesting a much higher surface area of

the coatings. A difference among the bulk materials was observed, where the highest value was obtained for the 12CoUF sample, while the values for the 12CoF and 12CoM samples are almost the same. The n values are consistent with heterogeneous current distribution all over the sample surface, being the lowest value for the polished HVAF sample. Accordingly, the resistance (R) increases as the admittance decreases, mainly for bulk materials, indicating that the surface processes attributed to R are more resistant for the 12CoM sample. For CGS coatings, it is well known that the top layers may be less compact than the bottom ones [43]; this means that the as-sprayed CGS coating essentially consists of a top layer with some porosity and a dense bottom layer, which explains the use of the CPE element instead of C to describe the electrical double layer capacitance. The n and the high admittance values are justified by the presence of a less compact top layer in the as-sprayed CGS coating. For the other samples, a capacitor may represent the electrical double layer with higher capacitance, as expected, for the HVAF coating, probably due to a higher active surface area. The capacitance values of the electrical double layer for the bulk materials are similar to those obtained for metals [44], and they decrease from 12CoUF to 12CoM while the charge transfer resistance increases, suggesting that this process is easier for 12CoUF than for 12CoM. This result seems to indicate that the 12CoUF sample is more active for cobalt dissolution than the other bulk materials, in agreement with the other electrochemical results, which can be related to the microstructure of this sample and probably to the smaller size of the WC particles.

To better understand the results, more details about the structure of the sample and especially about the evolution of the surface with the immersion time could be useful. The low $R_{ct}$ value for CGS coating can be attributed to a more effective galvanic effect around the WC particles together with potential higher residual stress/strain in cobalt caused by the spraying process. The higher $R_{ct}$ value obtained for polished HVAF samples can be related to the more resistive nature of the sample due to the presence of complex cobalt–tungsten carbides and, eventually, the lower exposed cobalt mainly in holes and pores. For further comprehension of CGS coating, a thicker and polished coating should be prepared and investigated.

## 5. Conclusions

CGS and HVAF WC-12 wt%Co coatings differ in their microstructural features and show a complex heterogeneity when compared to those of bulk materials. The solid-state nature of the CGS process implies a coating buildup mechanism based on particle deformation of impinging particles. This leads to low deposition efficiency, considerable carbide fracture, and internal residual stress/strain in cobalt binder, which is different from the partial or full melting of the cobalt phase in the HVAF coating where carbide dissolution is promoted. Together with surface roughness (polished/as-sprayed), all these features have been observed to directly affect the electrochemical corrosion performance.

The as-sprayed CGS coating presents electrochemical behavior similar to that of the bulk materials, probably due to its higher metallic character as compared to the HVAF coating. The EECs determined from EIS measurements for the as-sprayed CGS coatings, and all polished samples were two subcircuits in cascade. The as-sprayed CGS coating showed the lowest charge transfer resistance, followed by the bulk materials and the polished HVAF coating. The as-sprayed HVAF coating showed a very high resistive behavior.

Among the bulk materials, the charge transfer resistance increased in the order 12CoUF < 12CoF < 12CoM, and the electric double layer decreased in the same order.

**Author Contributions:** Conceptualization, N.C. and A.V.B.; methodology, O.L., S.C., S.H., S.D., H.K., R.J. and C.K.; validation, N.C., O.L. and A.V.B.; formal analysis, N.C., O.L., R.N.P, F.S.d.S. and A.V.B.; investigation, N.C., O.L., S.C., S.H., C.K., R.J. and F.S.d.S.; resources, E.T., S.C., S.H., S.D., H.K., C.K. and V.M.; data curation, A.V.B.; writing—original draft preparation, N.C., O.L. and A.V.B.; writing—review and editing, N.C., O.L., H.K., S.D., S.C., E.T. and A.V.B.; visualization, E.T.; supervision, N.C. All authors have read and agreed to the published version of the manuscript.

**Funding:** This research received no external funding.

**Institutional Review Board Statement:** Not applicable.

**Informed Consent Statement:** Not applicable.

**Data Availability Statement:** Not applicable.

**Acknowledgments:** The authors would like to acknowledge the Thermal Spray Center (CPT) in the University of Barcelona for the use of cold gas spray facilities to produce the CGS coatings.

**Conflicts of Interest:** The authors declare no conflict of interest.

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
