# Peer review of "Electrochemical Corrosion Characterization of Submicron WC-12Co Coatings Produced by CGS and HVAF Compared with Sintered Bulks"

_coatings, doi:10.3390/coatings12050620_

Round 1
Reviewer 1 Report
The current article intends to synthetize and compare HVAF and CGS coatings with bulk sintered WC-12Co samples regarding the microstructure and corrosion behaviour in neutral 3.5% NaCl solution. The topic of this manuscript is actual and interesting, it presents a great scientific importance, the paper is well-structured, though there are a few issues that need to be addressed towards the improvement of this work before publication. Some observations are summarized below:
- Please carefully proof-read spell check to eliminate grammatical and spelling errors like: line 58 ‘mainly amount of binder’, line 70 ‘which became as the industrial…’, line 80 ‘appears to be quite appealing’, line 84 ’is first aimed at comparing’,
- Line 53, line 55, line 73, line 83, etc.: References should be grouped like: [2-4], [5-9], [12-15], [17-19], etc.
- Line 21, line 157 – 3.56 wt. % NaCl – Here, I am not sure if there is a mistake. Usually, the corrosion investigations are performed in 3.5 wt. % NaCl aqueous solutions.
- It is not clear why did the authors choose the AA 7075-T6 alloy as a substrate material.
- Line 128: Although it is stated that the SEM microscope is coupled with an X-ray microanalysis system, I could not find any data regarding this investigation technique. It is strongly necessary to provide such data, to sustain some affirmations included also in the manuscript, regarding the oxidation of the samples’ surface (cobalt oxide layer formation) during the sample preparation (Line 237-239).
- The article contains several cross-references related to the figures included in text, some of them are referred as Figure and other as Authors should use a uniform terminology for the whole manuscript.
- Regarding the electrochemical investigations for the corrosion resistance assessment, how many repetition were performed for each specimen? Moreover, it is not clear if the potentiodynamic curves were measured on the same specimens which were previously tested for EIS (line 171-173). Please clarify this aspect.
- The XRD spectra of the CGS sample as well as bulk sintered specimens (Figure 6 a and b) depict only metallic cobalt in the structure, although the possibility of cobalt intermetallic phases formation with other elements it is not to be neglected.
- Figure 7 – there are two blue plots (probably blue and purple) that can be easily confused. I suggest picking another color for one of them to avoid misunderstanding.
- The corrosion behaviour was not analyzed in terms of corrosion current density, which is known to be an important parameter for corrosion resistance assessment. In my opinion, analyzing the results, there is a big difference between the HVAF as-sprayed and HVAF polished, regarding the corrosion current density (a 100 times higher corrosion current density for HVAF polished compared to the as-sprayed one). The authors should check this aspect, since the HVAF-as sprayed coating behaves totally different in the EIS measurements compared to the other investigated specimens.
- The sentence from line 226-228: Observing the cathodic branch one can see that almost all curves present a tendency to achieve a limiting current at more negative potentials, whereas the anodic branch shows the current density increasing continuously. The authors should clarify what does this affirmation imply.
- Line 301: For the first time in the manuscript, the presence of amorphous phases is mentioned. How are these phases formed and how are they evidenced in the XRD spectra, as I am not able to see any typical broad peak specific to the amorphous structures.
- Line 394-395: The authors affirm that the amount of oxides present on the surface is low. Is this affirmation sustained by some analysis?
- Table 4 is too large and is does not fit the manuscript’s template.
Author Response
Authors appreciate the reviewer's comments. Attached are the responses.

Reviewer 2 Report
WC-12 wt-%Co coatings are prepared through CGS and HVAF methods and their corrosion performance was tested using electrochemical impedance spectroscopy and potentiodynamic polarization studies. The authors claim that the as-sprayed CGS coating showed the lowest charge transfer resistance, followed by the bulk materials, and polished HVAF coating. The manuscript is well written and the results are discussed well. However, this cannot be published in the present form and there are few comments that the authors need to be addressed before publication.
- The authors claim that no such studies exist on the corrosion performance of WC-12Co coatings, but I have found many, and few of them are listed below.
https://www.webofscience.com/wos/woscc/full-record/WOS:000426701000034
https://www.worldscientific.com/doi/abs/10.1142/S0218625X18501147
- Please revise the introduction part based on the corrosion performance of previously reported WC-12Co coatings and highlight the novelty of your proposed work. It is good to have one paragraph about the corrosion performance of WC-12Co coatings.
- Why does the porosity differ between CGS and HVAF? What is the determining factor? Please discuss and include in manuscript.
- Figure 6b – index the small peak around 43.5 & 47 degrees.
- I couldn’t see any proof of coating thickness. Either show a cross-sectional SEM image or a profilometry graph. Also, the proof of FCC & HCP Co from XRD is not convincing. The author requested to justify the presence by TEM SAED pattern or any other related technique.
- The author claims that surface roughness affects corrosion performance. But the author didn’t provide the surface roughness data/profile.
- 3 – please address the presence of cracks and porosity within the coatings. Is it due to the optimized processing parameters? It would be better, if you could provide the low magnification images to see the growth morphology throughout coating from the substrate interface.
- Fig 8 & 9 combined to a single image, if possible. Also, Fig.7 – please increase the line width. It is difficult to see and differentiate. And Fig.9 fitted data is missing.
- In XRD, JCPDS card number is missing. Please discuss about lattice parameter values and crystallite size information.
- No such proof was provided for the formation of WC-12 wt-%Co coating.
Author Response

(The authors gave the same response as above.)

Round 2
Reviewer 1 Report
The manuscript can be accepted in the current form.
Reviewer 2 Report
Thank you for submitting the revised manuscript.